# All-Dielectric Meta-Surface for Multispectral Photography by Theta Modulation

**DOI:** 10.3390/nano10020369

**Published:** 2020-02-20

**Authors:** Meng Xiang, Dengfeng Kuang, Weichao Kong, Zengxin Huang, Panchun Gu, Wenshuang Li

**Affiliations:** Tianjin Key Laboratory of Micro-scale Optical Information Science and Technology, Institute of Modern Optics, Nankai University, Tianjin 300350, China; 2120170264@mail.nankai.edu.cn (M.X.); 422977786@qq.com (W.K.); 2120180243@mail.nankai.edu.cn (Z.H.); 1120170098@mail.nankai.edu.cn (P.G.); 15822274501@163.com (W.L.)

**Keywords:** meta-surface, theta modulation, visible light, geometric phase

## Abstract

The traditional theta modulator encodes input information by superimposing Ronchi sub-gratings, which is extremely easy to cause spatial channel overlap that results in bands mixing. In this case, we present an all-dielectric theta modulation meta-surface with a new encoding method, which separates red, green, blue, and achromatic spatial channels on the focal plane. The meta-surface ensures that the positions of focal points are relatively consistent while focusing energy into the sub-wavelength regions. Our study offers a way to facilitate device miniaturization and system integration, which may have an important application in compact multispectral photography only with one detector.

## 1. Introduction

Multispectral photography obtains the grayscale images of each band of the detection target. The grayscale difference of each band is used to superimpose to form various false color images to determine the characteristics of the target. Because the reflection and radiation characteristics of substances in each band are different, multispectral photography is utilized as an effective tool to measure and analyze targets noninvasively. A traditional approach of acquiring multi-spectral images of several bands takes several exposures of the target one at a time with one exposure per spectral band. Each spectral band is obtained by switchable filtering elements like tunable filters [1] and tunable illumination [2]. However, this strategy must employ complex and precise optical components, which results in a high cost and poor portability. The time and space mismatch of the obtained signals of various bands makes the images registration difficult. Another straightforward form is to use an optical beam splitter (such as prism) to split light into several spectral bands in which each is captured by a detector [3]. However, such a camera structure is composed of multiple detectors and is considered bulky and expensive. A more serious obstacle is that splitting the incident light into several parts will quickly reduce the light energy of each spectral band to unacceptable levels. Theta modulation originated from the famous Abbe-Porter experiment and was applied to produce color images from black and white film [4]. Based on the theta modulation technique, optical multiple storage [5], color mixture technology [6], optical logic processing [7], and optical encryption technology [8] are subsequently demonstrated. Represented by the tricolor grating [9,10,11,12], a traditional theta modulator is formed by superimposing Ronchi sub-gratings at three different directions. A tricolor grating can transform the original scene into a grating-encoded image, which means that spectral bands are subsampled in space and several types of spectral samples are interleaved in a detector. Then, these spectral samples are subjected to appropriate spatial filtering to obtain multispectral images at the same time. Combined with theta modulation, a multispectral photography system consisting of a multiband theta modulator and a single detector can replace the traditional approaches and enjoys the advantages of simplified hardware architecture, lower cost, and higher adaptability. However, the spatially overlapping portions of the tricolor grating cause various bands mixing, and cross-terms may appear between adjacent bands on the spectrum, which limits the bandwidth and accuracy of multispectral information. Moreover, there are also two problems in fabricating a tricolor grating: (1) prime grating should have a suitable spatial frequency and high contrast, and (2) the tricolor grating should fit the best orientation to avoid overlapping of diffraction orders and producing the Morie fringe. Therefore, great uniformity and perfect processing are required to ensure high definition of color images. 

Meta-surfaces are planar optical elements that hold promise for overcoming the limitations of conventional refractive and diffractive optics due to their flexibilities of the control phase, amplitude, and polarization by sub-wavelength scatterings [13,14,15,16,17]. Meta-surfaces lead to generalize the classic laws of reflection and refraction [18] and Brewster effect [19] on a flat surface. Furthermore, meta-surfaces can change the way that traditional optical components rely primarily on the inherent properties of materials to achieve the desired effects [20], which leads to the bulk and complicated systems. Compared to diffractive elements, the sub-wavelength structures constituting meta-surfaces are free of high-order undesired diffraction from the propagation phase accumulation, and could provide more accurate and efficient phase control than binary phase or amplitude zone plates [21,22]. Meta-surfaces are superior to refractive elements because they are lightweight, have a compact size, and offer more design freedoms. The theoretical model study of meta-surfaces becomes a new research direction [23]. Correspondingly, meta-surfaces can realize a series of applications on flat lenses [24,25,26], color filters [27,28,29], meta-holograms [30,31], compact spectrometers [32], and active tuning of the spectral position [33]. 

In order to implement a highly efficient meta-surface, the complex refractive index (*n = n + ik*) of dielectric materials should meet two conditions: high refractive index and low loss [34]. In this scenario, we propose theta modulation titanium dioxide (TiO_2_) meta-surface [35] to achieve high operation efficiency at a visible light range by introducing the Pancharatnam-Berry (PB) phase [36] to design the meta-surface and to improve the grating frequency to match the size of the photosensitive pixels of the CCD/CMOS detector. The theta modulation meta-surface is capable of simultaneously focusing light of different wavelengths to different spatial locations on the same focal plane to perform red, green, blue, and achromatic spatial channels, respectively, and greatly solving the bands mixing problem of traditional tricolor grating caused by spatial channel overlapping.

## 2. Design Methods 

The traditional theta modulator is represented by a tricolor grating [12], which can be regarded as a superposition of Ronchi sub-gratings in three different directions, formed by exposing the three bands of visible red, green, and blue in three different directions on the color film, as shown in Figure 1a. In this tricolor grating, the spectral transmittance of three-direction channels partially overlaps in some wavelength ranges. The light *λ_gb_* in the wavelength range of 480 nm to 530 nm can pass through both the blue channel and the green channel simultaneously, which generates an additional grating of the blue-green channel, and a similar phenomenon occurs in the light *λ_r_*_g_ in the wavelength range of 580 nm to 630 nm. The intensity distribution of the image encoded by the conventional multi-band theta modulator can be expressed as:(1)$$\begin{array}{l} f_{t}(x,y)\\ =f_{r}(x,y)P_{r}(x_{r})+f_{g}(x,y)P_{g}(x_{g})+f_{b}(x,y)P_{b}(x_{b})+f_{rg}(x,y)P_{rg}(x_{r},x_{g})+f_{gb}(x,y)P_{gb}(x_{g},x_{b}) \end{array}$$
where *f* (*x*, *y*) is the component of the incident signal function in different bands, *P* (*x*, *y*) represents the spectral transmittance of the theta modulator, *x_r_*, *x_g_*, and *x_b_* are the orientations of the red (R), green (G), and blue (B) gratings, respectively. Equation (1) indicates that the incident information of different wavelength bands modulated is recorded on one detector at the same time. Then, the corresponding Fourier spectrum can be expressed as:(2)$$\begin{array}{l} F^{’}(u,v)=F(u,v)+F_{rg}(u,v)⊗\sum_{n_{r}=-\infty }^{\infty }\sum_{n_{g}=-\infty }^{\infty }[\delta (u_{r}+n_{r}u_{0})\sin c(n_{r}u_{0})\delta (u_{g}+n_{g}u_{0})\sin c(n_{g}u_{0})]\\ +F_{gb}(u,v)⊗\sum_{n_{g}=-\infty }^{\infty }\sum_{n_{b}=-\infty }^{\infty }[\delta (u_{g}+n_{g}u_{0})\sin c(n_{g}u_{0})\delta (u_{b}+n_{b}u_{0})\sin c(n_{b}u_{0})] \end{array}$$
where *F*(*u*, *v*) is the encoded Fourier spectrum of the signal function of the red, green, and blue bands. The grating frequency is *u_0_*, and *u_r_*, *u_g_*, and *u_b_* are the orientations of the Fourier spectrum corresponding to the R, G, and B gratings, respectively. The second and third terms in the equation are Fourier spectral terms of *f_rg_* (*x*, *y*) and *f_gb_* (*x*, *y*) after encoding by each additional grating, *n* = ±1, ±2.

Since the first-order spectrum of the red channel is close to the first-order spectrum formed by the additional grating *P_gb_*, the distance is *d_r_* = ($\sqrt{2}$ − 1) *u_0_* ≈ 0.41 *u_0_*. In spectral filtering, the first-order spectrum formed by the additional grating *P_gb_* is particularly easy to mix into the first-order spectrum of the red channel to cause the undesirable cross terms. The same problem can also occur between the blue channel primary spectrum and the primary spectrum of the additional grating *P_rg_*, and *d_b_* = *d_r_*. In the application of multispectral photography, it is difficult to obtain real information from a single band, which affects the bandwidth and accuracy of multispectral photography.

In combination with the photosensitive pixels of current detectors, which are arranged in a matrix of two-dimensional blocks, we design a new encoding method of theta modulator in which there are no spatial overlaps in the red, green, and blue bands, as shown on the left side of Figure 1b. The right side of Figure 1b shows a schematic of the basic unit consisting of red (R), green (G), blue (B), and achromatic (A) areas. The A area contains information on the three-color bands. According to Nyquist’s law of sampling, for this new method, each basic unit period needs to contain two or more data points. In order to increase the spatial resolution, each basic unit period contains two data points in our design. Thus, if the cell width of the area array CCD/CMOS detector is *w*, the periods of the three basic units of red, green, and blue of the new encoding theta modulator are 2*w*, 2*w*, and 2$\sqrt{2}$*w*, respectively. The photosensitive area of the CCD/CMOS detector is *S* × *S*. The spectral transmittance of this theta modulator can be expressed as:(3)Pt(x)=Pr(xr)+Pg(xg)+Pb(xb)={rect[xr(w)]comb[xr(2w)]}rect(xrS)+{rect[xg(w)]comb[xg(2w)]}rect(xgS)+{rect[xb(2w)]comb[xb(22w)]}rect(xb2S)

After Fourier transform, the corresponding spectrum is obtained as:(4)$$\begin{array}{l} F_{T}\left(u,v\right)=2Sw^{2}comb\left(2wu_{r}\right)\sin c\left(wu_{r}\right)\sin c\left(Su_{r}\right)+2Sw^{2}comb\left(2wu_{g}\right)\sin c\left(wu_{g}\right)\sin c\left(Su_{g}\right)\\ +2\sqrt{2}Sw^{2}comb\left(2\sqrt{2}wu_{b}\right)\sin c\left(2\sqrt{2}wu_{b}\right)\sin c\left(\sqrt{2}Su_{b}\right) \end{array}$$

For the red, green, and blue color channels, the cutoff frequencies are 1/(2*w*), 1/(2*w*), and 1/($2\sqrt{2}$*w*), which coincide with the position of the first-order spectrum. The spectrum of the three channel signals does not overlap at all, and the true information of each band can be strictly obtained.

With the development of semiconductor technology, the size of photosensitive pixels of CCD/CMOS detectors is generally 4~8 μm. It not only requires the size of each area of the redesigned theta modulator to match the size of the photosensitive pixels of the CCD/CMOS detector, but their position also corresponds to each other. Therefore, the meta-surface becomes our best choice to construct the theta modulator because it can concentrate the energy and achieve the sub-wavelength focusing. As shown in Figure 1c, the R, G, and B areas work as three initial chromatic meta-lenses, and focus red, green, and blue light to different positions on the same focal plane, respectively. The A area can be regarded as an achromatic metalens focusing the three colors with the same focal length as the R, G, and B areas.

In this case, we implement a dielectric meta-surface consisting of TiO_2_ nano-pillars with only three sizes with sub-wavelength spacing to manipulate the phases of the red, green, and blue wavelengths. The TiO_2_ nano-pillar on the silica (SiO_2_) substrate is shown in Figure 2a as the basic unit for constructing the theta modulation meta-surface. TiO_2_ material is modeled with a refractive index (*n*) and an extinction coefficient (*k*) and fitted with a universal dispersion model [37] in our numerical simulation. When an incident circularly polarized beam vertically passes through nano-pillars, the transmitted beam consists of the original polarization portion without a phase shift and the converted cross polarization portion with an additional phase shift. For the cross polarization transmitted waves, the additional phase shift is *± 2α* (*α* is the orientation angle of nano-pillar) where the “±” sign is determined by the helicity of the incident light. Due to the birefringence caused by the asymmetric cross section of the nano-pillars, coupled with the wavelength independence of the PB phase, the geometric parameters of the nano-pillars can be appropriately designed to maximize the polarization conversion efficiency. The polarization conversion efficiencies are calculated as the ratio of transmitted optical power with cross polarization to the total incident power. For simplifying the calculation, we set the height of the TiO_2_ nano-pillar as *h =* 600 nm and the depth of the SiO_2_ substrate as *d* = 200 nm. Square substrate lattices with different periods (*p*) are selected for the optimized and designed simulations at different visible wavelengths with *p* = 430 nm for red (*λ* = 700 nm), *p* = 325 nm for green (*λ* = 513 nm), and *p* = 200 nm for blue (*λ* = 405 nm). The period needs to be smaller than the main working wavelength to avoid diffraction effects, but also large enough to avoid near-field strong interactions between two adjacent nano-pillars. As shown in Figure 2b–d, the polarization conversion efficiencies could be expressed as functions of the length (*l*) and width (*w*) of the rectangular nano-pillars for incident circular polarized light at red, green, and blue, respectively. Clearly, for a given condition, multiple nano-pillar size options may be encountered. Satisfactory polarization conversion efficiency (greater than 80%) appears in multiple size choices, which means that the length and width of the nano-pillars are highly available. Therefore, the rectangular nano-pillars provide more flexibility in the choice of the dimensions, which relaxes the resolution requirement and eases the fabrication constraints. Then, we choose the nano-pillars with appropriate lengths and widths as the basic unit of the meta-surface, as displayed in Table 1.

In order to verify the working performance of the selected nano-pillars, we perform a full visible spectrum sweep of the three-size nano-pillars with their substrate lattice periods. The polarization conversion efficiencies are shown in Figure 3a with the sizes of nano-pillars displayed in Table 1. By controlling the local orientation of the longer axis of nano-pillars between 0 to π, the phase shift pickups achieved the entire 0 to 2π range.

As long as the incident plane wave is converted into a spherical wave, we can realize a meta-surface as planar lens. To function like a spherical lens, the phase profile of meta-lens needs to follow the equation below [38].
(5)$$\Phi (x,y)=\pm \frac{2\pi }{\lambda }[\sqrt{x^{2}+y^{2}+f^{2}}-f]$$
where *λ* is the design wavelength, *x* and *y* are the coordinates of each nano-pillar, and *f* is the focal length. Note that the “+” and “−” signs in Equation (5) corresponds to convex and concave state, respectively, for incident right circular polarization (RCP) and left circular polarization (LCP) light beam. For an arbitrary focus *F*(*x_f_* , *y_f_* , *z_f_* ), the phase profile for the meta-surface, which is able to converge the light into free space can be written as:(6)$$\Phi (x,y)=\pm \frac{2\pi }{\lambda }[\sqrt{{\left(x-x_{f}\right)}^{2}+{\left(y-y_{f}\right)}^{2}+{\left(z-z_{f}\right)}^{2}}-z_{f}]$$

Thus, it is easy to design three distinct meta-lenses with corresponding parameters for red (*λ* = 700 nm), green (*λ* = 513 nm), and blue (*λ* = 405 nm) at R, G, and B area. For the achromatic meta-lens at the A area, we multiplex the nano-pillars in response to red, green, and blue light into a complex cell that provides multi-channel full phase manipulations at R, G, and B wavelengths. In order to facilitate the arrangement of the nano-pillars, the complex cells need to be square pixels, which means that the R, G, and B nano-pillars are arranged with the same period as *p* = 430 nm. We calculate the polarization conversion efficiencies in this issue, as shown in Figure 3b. Comparing Figure 3a,b, there is an inevitable decrease in polarization conversion efficiency of blue and green nano-pillars due to a decrease in an effective area across the period. Since the largest decrease in the polarization conversion efficiency of a blue nano-pillar, we add another blue nano-pillar to one complex cell, as shown in the illustration of Figure 3b.

## 3. Result and Discussion

Following the above strategy, an achromatic meta-lens is demonstrated numerically with NA = 0.718 at red (*λ* = 700nm), green (*λ* = 513 nm), and blue (*λ* = 405 nm). Figure 4 shows the normalized electric field intensity distributions in the *xz* plane for an achromatic focusing (a) and an initial chromatic focusing (b) under incident LCP light at three wavelengths. It is clearly illustrated in Figure 4a that the focal lengths are adjusted to the same position at three wavelengths, whereas a large chromatic dispersion exists in Figure 4b. The variance of the focal lengths is only 0.30 μm (*f_λ_**_=_**_700 nm_* - *f _λ_**_=_**_513 nm_* = 10.35–10.05 μm) for the achromatic meta-lens. For the chromatic case, the focal length at *λ* = 405 nm is 7.21 μm (*f_λ=700 nm_* - *f_λ=405 nm_* = 14.02–6.81μm) larger than that at *λ* = 700 nm.

Next, we integrate the red, green, and blue meta-lenses and the achromatic meta-lens into a 20.64 μm × 20.64 μm square to construct the theta modulation meta-surface. Four areas are arranged equally, as shown in Figure 5a, and the design parameters are depicted in Table 2. Each area focuses light to their own center with *f* = 10.00 μm.

We illuminate the theta modulation meta-surface with different incident lights and calculate the electric field intensity distribution. It is calculated that positions of red, green, and blue focus are R (5.17 μm, 15.49 μm, 10.05 μm), G (15.51 μm, 5.15 μm, 10.08 μm), and B (15.47 μm, 15.47 μm, 10.20 μm). The positions of achromatic focus are located at A *_λ = 700 nm_* (5.17 μm, 5.20 μm, 10.28 μm), A *_λ = 513 nm_* (5.15 μm, 5.19 μm, 9.00 μm), A *_λ = 405 nm_* (5.18 μm, 5.15 μm, 10.05 μm). We select the *xz* plane with *y* = 15.48 μm (Figure 5b,e,h), the *xz* plane with *y* = 5.16 μm (Figure 5e–g), and the *xy* plane with *z* = 10.00 μm (Figure 5h–j) to observe the electric field distributions when red, green, and blue lights are illuminated individually. The full widths at half maximum (FWHMs) of red, green, and blue focus spots are 636 nm, 509 nm, and 403 nm, respectively. For the achromatic focus, the FWHMs of the incident at distinct wavelengths are 665 nm, 520 nm, and 426 nm, respectively. The focusing efficiency can be defined as the portion of the incident light that passes through a circular iris in the focal plane with a radius equal to three times the FWHM spot size, which is defined as the total power in the desired focal spot divided by the total incident power [39]. It is calculated by the focusing efficiencies of R, G, B, and A area are 9.98%, 17.40%, 30.47%, and 5.66%, respectively. Although the red, green, and blue lights are spatially separated according to the design, the achromatic focusing efficiency is much lower than that of other colors. In this case, the lowest efficiency is derived from achromatic focusing, which is due to the complex cell and results in a reduced number of effectively modulated nano-pillars across the meta-surface.

Considering the problem of low efficiency and reduced effective nano-pillars of the achromatic meta-lens in the A area, we design the four areas as an unequal proportion, as depicted in Table 3.

The total area of the meta-surface still remains 20.64 μm × 20.64 μm, and the design focal length is also 10.00 μm. The area of A, R, G, and B is occupying 4/9, 2/9, 2/9, and 1/9 of the total meta-surface, respectively, as shown in Figure 6a.

In order to verify the modulation effect of the theta modulation meta-surface, we illuminate it with different incident lights and calculate the electric field intensity distributions. It is calculated that positions of red, green, and blue focus are R (6.89 μm, 17.36 μm, 10.09 μm), G (17.23 μm, 6.88 μm, 10.13 μm), and B (17.18 μm, 17.19 μm, 10.28 μm). The positions of achromatic focus are located at A *_λ = 700 nm_* (6.89 μm, 6.89 μm, 10.28μm), A *_λ = 513 nm_* (6.90 μm, 6.88 μm, 10.18 μm), and A *_λ = 405 nm_* (6.88 μm, 6.88 μm, 10.28 μm). All of the positions of focus spots show a good consistency with our prediction. The electric field intensity distributions of *xz* plane at *y* = 17.20 μm when the theta modulation meta-surface is illuminated by red, green, and blue light separately, which are shown in Figure 6b,e,h. Corresponding results at *y* = 6.88 μm are shown in Figure 6c,f,i, respectively. Then, we select the *xy* plane with *z* = 10.28 μm (marked by a white dash line in Figure 6) as the focal plane to analyze the modulation effect. The images of the focal plane are depicted in Figure 6d,g,j. Meanwhile, the normalized intensity distributions along the horizontal (*x*) line through the center of the focus are noted. The FWHMs of red, green, and blue focus spots are 680 nm, 592 nm, and 554 nm, respectively. For the achromatic focus, the FWHMs of the incident at distinct wavelengths are 576 nm, 405 nm, and 342 nm, respectively. The focusing efficiencies of R, G, B, and A areas are 9.58%, 17.03%, 15.35%, and 9.83%, respectively. Compared to the case where the four areas are arranged equally, the energy distribution of each focus of the theta modulation meta-surface is more balanced. Although the areas of the four regions are unequal, all meta-lenses we designed collect the incident light in a sub-wavelength region, and the positions of the respective focal points are always relatively consistent, so that the large-area array becomes a theta modulation meta-surface with our new encoding method to avoid overlapping of various spatial channels.

## 4. Conclusions

In conclusion, we demonstrated an all-dielectric theta modulation meta-surface with a new encoding method by utilizing a series of TiO_2_ nano-pillars, which is capable of modulating an individual wavelength to correspondingly desired positions without spatial channel overlaps. Compared to the traditional tricolor grating, the new encoding method improves the grating frequency, so that the size and position of each region of the theta modulator are in one-to-one correspondence with the photosensitive pixels of the CCD/CMOS detector by solving the problem of bands mixing. At the same time, applying our lightweight design where the thickness is only 800 nm close to the wavelength to multispectral photography, which is a streamlined system consisting of a meta-surface with a new encoding method and a single detector can greatly reduce the number of precision optical components and detectors. This results in lower cost and higher portability. The information obtained in multiple bands is matched in time and space, which can effectively shorten the time of subsequent information processing. Our study offers a feasible way for designing integrated optical devices, which may find applications in compact imaging, optical coupling, and multispectral recognition technology.

## Figures and Tables

**Figure 1 nanomaterials-10-00369-f001:**
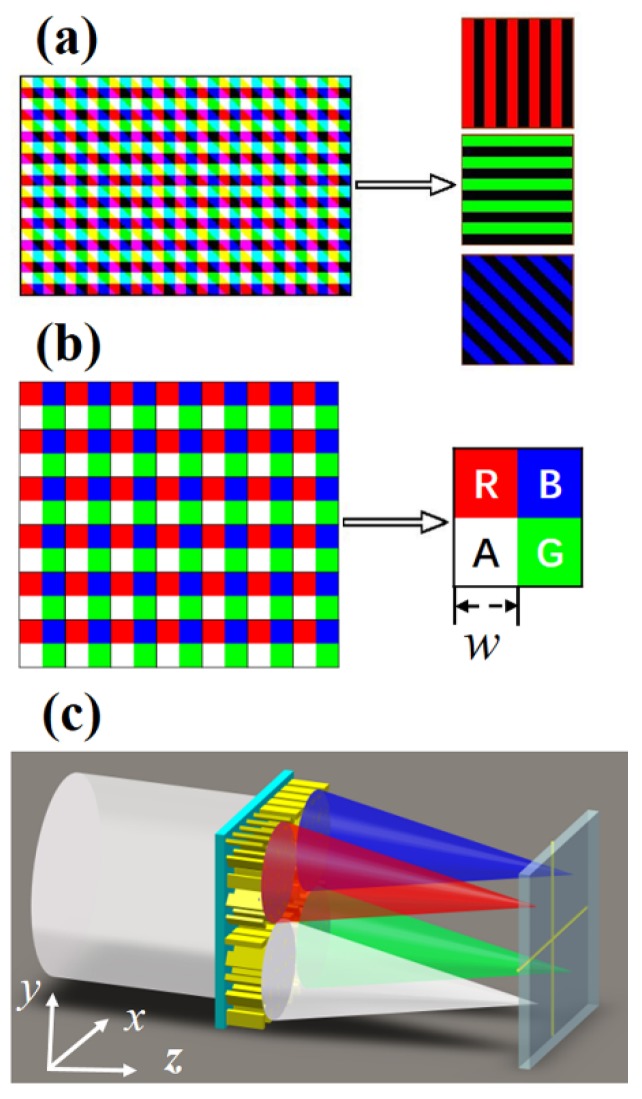
(**a**) Schematic illustration of traditional tricolor grating. (**b**) Schematic illustration of the encoding method with a multi-wavelength theta modulation meta-surface. (**c**) Schematic side view of the multi-wavelength theta modulation meta-surface.

**Figure 2 nanomaterials-10-00369-f002:**
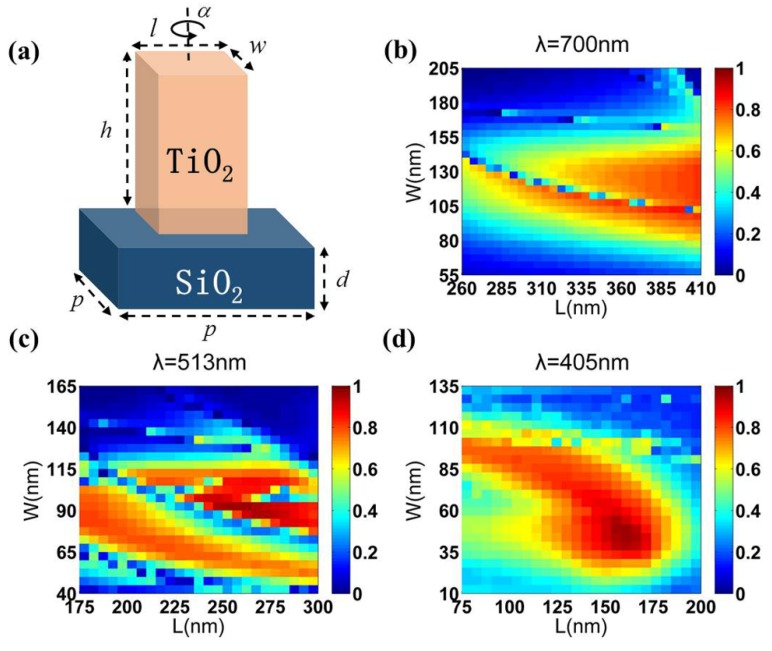
(**a**) The sketch of a TiO_2_ nano-pillar on a SiO_2_ substrate. (**b**–**d**) The polarization conversion efficiencies of the rectangular nano-pillars as a function of l and w for normal incidence of the circular polarized light at red, green, and blue, respectively.

**Figure 3 nanomaterials-10-00369-f003:**
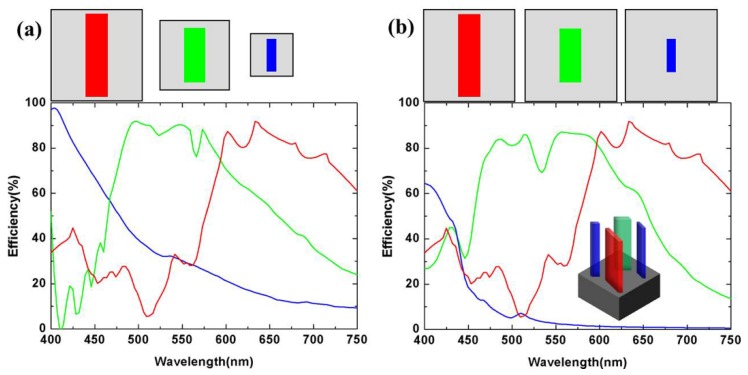
(**a**) The top is the diagram of top view of three nano-pillars (the corresponding sizes are shown in Table 1. The bottom is the simulated polarization conversion efficiency as a function of the wavelength). (**b**) The top is the diagram of the top view of three nano-pillars with the same period as *p* = 430 nm. The bottom is the simulated polarization conversion efficiency as a function of wavelength. The illustration is a schematic of a complex cell.

**Figure 4 nanomaterials-10-00369-f004:**
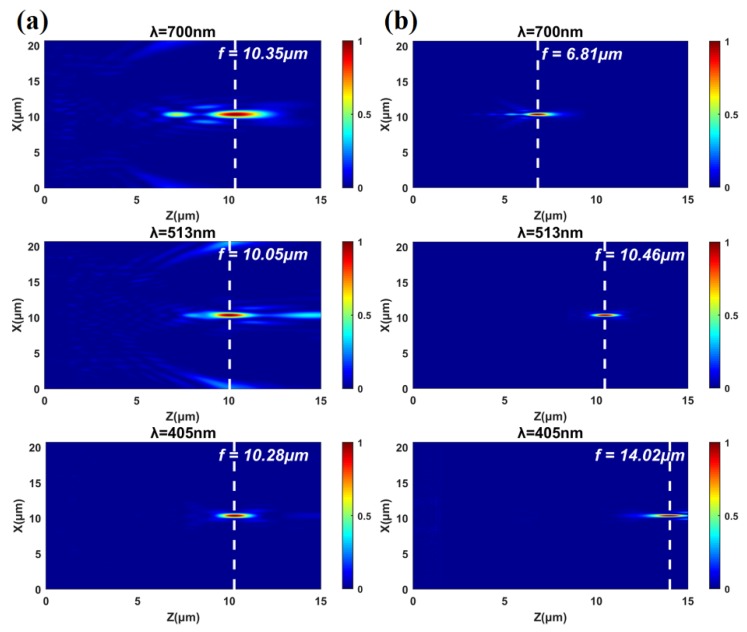
Normalized electric field intensity distributions in the *xz* plane for (**a**) an achromatic focusing and (**b**) an initial chromatic focusing.

**Figure 5 nanomaterials-10-00369-f005:**
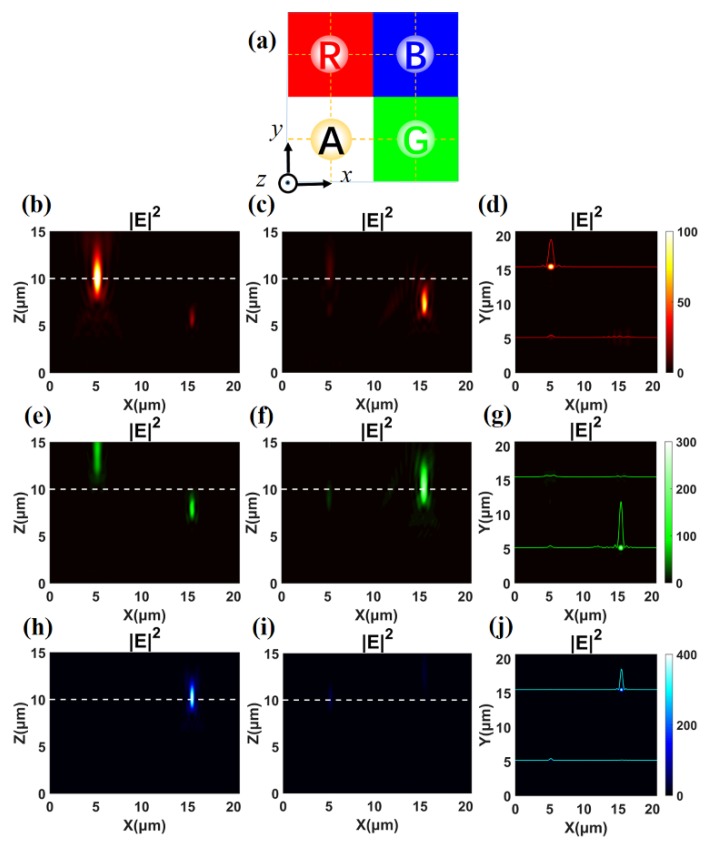
(**a**) Schematic diagram of the theta modulation meta-surface with four equal areas. (**b**–**d**) Electric field intensity profiles in the focal region of the *xz* plane with *y* = 15.48 μm, *xz* plane with *y* = 5.16 μm, and *xy* plane *z* = 10.00 μm, respectively, at *λ* = 700 nm. The white dash lines mark the position of *z* = 10.00 μm in the *xz* planes. The normalized intensity distributions along the horizontal (*x*) line through the center of the focus are noted. (**e**–**g**), (**h**–**j**) Corresponding analysis of panels (**b**–**d**) for a meta-surface at *λ* = 513 nm and *λ* = 405 nm.

**Figure 6 nanomaterials-10-00369-f006:**
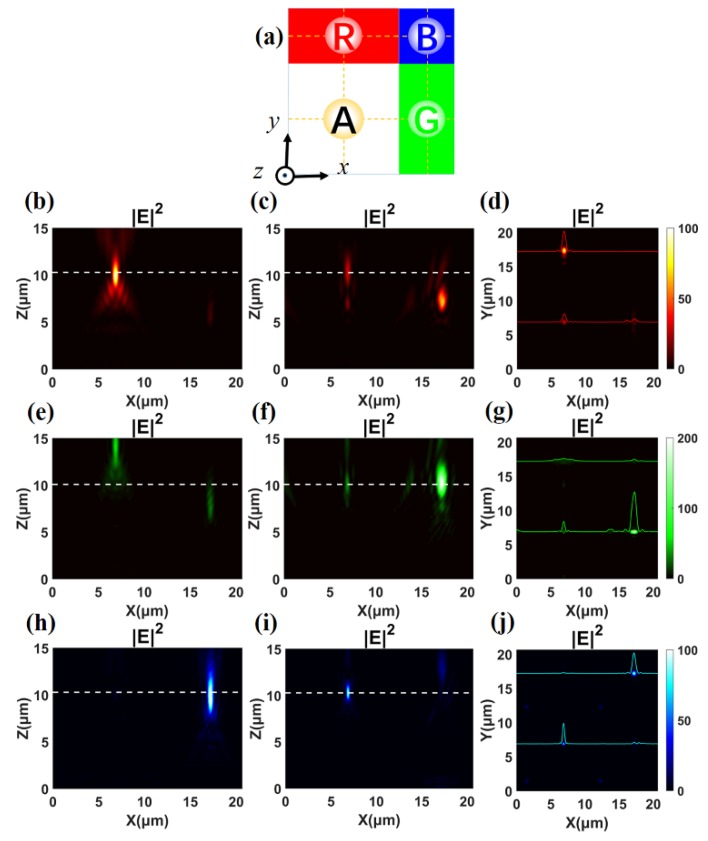
(**a**) Schematic diagram of the theta modulation meta-surface with four unequal areas. (**b**–**d**) Electric field intensity profiles in the focal region of *xz* plane with *y* = 17.20 μm, *xz* plane with *y* = 6.88 μm, and *xy* plane *z* = 10.28 μm, respectively, at *λ* = 700 nm. (**e**–**g**), (**h**–**j**) Corresponding analysis of panels (b–d) for a meta-surface at *λ* = 513 nm and *λ* = 405 nm.

**Table 1 nanomaterials-10-00369-t001:** Geometric sizes of three selected nano-pillars.

*λ _incident_*	700 nm	513 nm	405 nm
***p*** (nm)	430	325	200
***l*** (nm)	385	250	150
***w*** (nm)	100	95	40

**Table 2 nanomaterials-10-00369-t002:** Design parameters of four areas with equal proportions.

	Area (μm^2^)	Coordinates of the Focal Point (μm)
**A**	10.32 × 10.32	(5.16, 5.16, 10.00)
**R**	10.32 × 10.32	(5.16, 15.48, 10.00)
**G**	10.32 × 10.32	(15.48, 5.16, 10.00)
**B**	10.32 × 10.32	(15.48, 15.48, 10.00)

**Table 3 nanomaterials-10-00369-t003:** Design parameters of four areas with unequal proportions.

	Area (μm^2^)	Coordinates of the Focal Point (μm)
**A**	13.76 × 13.76	(6.88, 6.88, 10.00)
**R**	13.76 × 6.88	(6.88, 17.20, 10.00)
**G**	13.76 × 6.88	(17.20, 6.88, 10.00)
**B**	6.88 × 6.88	(17.20, 17.20, 10.00)

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
