# Peer review of "All-Dielectric Meta-Surface for Multispectral Photography by Theta Modulation"

_nanomaterials, 2020, doi:10.3390/nano10020369_

Round 1
Reviewer 1 Report
The manuscript titled "All-dielectric metasurface for multispectral photography by theta modulation" by Meng Xiang, Dengfeng Kuang, Weichao Kong, Zengxin Huang, Panchun Gu and Wenshuang Li deals with focusing and filtering of light using metasurfaces, which is an interesting topic in recent optical research. I found the subject of investigation actual for the state-of-the-art optics of metasurfaces. I think, the paper is usefull and gives some contribution to understanding construction of metasurface based beam focusers. But there are some issues that have to be addressed before publication.
The title. I don’t understand relation between multispectral photography and results in the work. Multispectral photography involves focusing of continues spectrum of light, but only discrete spectrum is investigated in the work (three wavelengths). There are no explanation about relief height of the metasurface. It is expected that for three considered wavelengths nano-pillar heights will be different, which will lead to problem in fabrication of the combined metalens. For example, the influence of the relief height on the metasurface quality is illustrated in [Opt. Express, 25: 8158 (2017); doi.org/10.1364/OE.25.008158]. Please, provider some explanation and data about relief height for different wavelengths. The inset with one combined pixel in Fig. 2d seems to be not real, but if it is, it should be discussed, why two blue cells have nano-pillars situated with different angles in the one pixel. Why when you use combined pixel (Fig. 2d), the efficiency stays almost the same, as when you use uniform meta surface (Fig. 2c)? It seems to be not real. For example, for green light, it is more expected, that the efficiency will decrease in more than 4 times because of decreasing in 4 times of the surface square used by green cells (1 of 4) and using of non-optimal greater common period of cells.
Reviewer 2 Report
The submitted manuscript deals with the design of an all-dielectric metasurface to be used in multispectral holography. Authors argue that, compared to a conventional tricolor grating, the proposed metasurface is able to reduce the superposition between different channels and, thus, to improve the accuracy of the multispectral photography system.
All-dielectric metasurfaces are a hot topic in the electromagnetic and optics community because of their multiple advantages over conventional solutions. There are many works about this topic but, to my knowledge, the idea discussed in the manuscript is relatively new and interesting. Results look promising, but some points should be rigorously addressed before publication may be finally recommended:
- It is not clear to me how the different unit-cells of the metasurface have been designed. Have authors employed an analytical model (at least for a first rough design) or, conversely, the unit-cells have been designed with numerical simulations?
- While the potential advantages of all-dielectric metasurfaces for multispectral photography are clearly discussed, I do not see a comparison of the final performances between conventional system and metasurface-based systems. This is important to appreciate, quantitatively, the advantages returned by the dielectric metasurfaces in realistic scenarios. In fact, having a look to the polarization conversion efficiency in Fig. 2, I see a significant spectral superposition between the different unit-cells, whose response is not as sharp as desired.
- How the Ti02 material has been modelled in the numerical simulation? Please, comment on the model used to describe the permittivity and the refractive index of this high-index dielectric.
- Please, add the size of the final designed elements in a table. Could you please comment on the robustness of this device to fabrication inaccuracies, which may be important for non-symmetric shapes with nanometer size.
- Please, clarify the differences between Fig. 2(c) Fig. 2(d).
- I think that some important works about the properties of all-dielectric metasurfaces should be included in the reference list. Before, some of them:
“Wave manipulation with designer dielectric metasurfaces,” Opt. Lett., vol. 39, pp. 6285–6288, 2014.
“Active tuning of all-dielectric metasurfaces,” ACS Nano, vol. 9, no. 4, pp. 4308–4315, 2015.
"Optically resonant dielectric nanostructures," Science, vol. 354, 2472, 2016.
“Surface Impedance Modeling of All-dielectric Metasurface,” IEEE Transactions on Antennas and Propagation, in press, doi: https://doi.org/10.1109/TAP.2019.2951521.
Round 2
Reviewer 2 Report
Thanks for replying to my comments.